# DR5-Selective TRAIL Variant DR5-B Functionalized with Tumor-Penetrating iRGD Peptide for Enhanced Antitumor Activity against Glioblastoma

**DOI:** 10.3390/ijms232012687

**Published:** 2022-10-21

**Authors:** Anne V. Yagolovich, Alina A. Isakova, Artem A. Artykov, Yekaterina V. Vorontsova, Diana V. Mazur, Nadezhda V. Antipova, Marat S. Pavlyukov, Mikhail I. Shakhparonov, Anastasia M. Gileva, Elena A. Markvicheva, Ekaterina A. Plotnikova, Andrey A. Pankratov, Mikhail P. Kirpichnikov, Marine E. Gasparian, Dmitry A. Dolgikh

**Affiliations:** 1Shemyakin-Ovchinnikov Institute of Bioorganic Chemistry RAS, 117997 Moscow, Russia; alina.labbio@gmail.com (A.A.I.); art.al.artykov@gmail.com (A.A.A.); kate.voronts09@gmail.com (Y.V.V.); diana2000.11@gmail.com (D.V.M.); nadine.antipova@gmail.com (N.V.A.); marat.pav@mail.ru (M.S.P.); shakhparonov@gmail.com (M.I.S.); sumina.anastasia@mail.ru (A.M.G.); lemarkv@hotmail.com (E.A.M.); kirpichnikov@inbox.ru (M.P.K.); marine_gasparian@yahoo.com (M.E.G.); dolgikh@nmr.ru (D.A.D.); 2Faculty of Biology, Lomonosov Moscow State University, 119192 Moscow, Russia; 3Manebio LLC, 115280 Moscow, Russia; 4National Medical Research Radiological Centre of the Ministry of Health of the Russian Federation, P.A. Hertsen Moscow Oncology Research Institute, 125284 Moscow, Russia; plotnikovaekaterina62@gmail.com (E.A.P.); andreimnioi@yandex.ru (A.A.P.)

**Keywords:** TRAIL, DR5 receptor, DR5-B, bispecific fusion protein, iRGD, tumor-penetrating peptide, glioblastoma

## Abstract

TRAIL (TNF-related apoptosis-inducing ligand) and its derivatives are potentials for anticancer therapy due to the selective induction of apoptosis in tumor cells upon binding to death receptors DR4 or DR5. Previously, we generated a DR5-selective TRAIL mutant variant DR5-B overcoming receptor-dependent resistance of tumor cells to TRAIL. In the current study, we improved the antitumor activity of DR5-B by fusion with a tumor-homing iRGD peptide, which is known to enhance the drug penetration into tumor tissues. The obtained bispecific fusion protein DR5-B-iRGD exhibited dual affinity for DR5 and integrin αvβ3 receptors. DR5-B-iRGD penetrated into U-87 tumor spheroids faster than DR5-B and demonstrated an enhanced antitumor effect in human glioblastoma cell lines T98G and U-87, as well as in primary patient-derived glioblastoma neurospheres in vitro. Additionally, DR5-B-iRGD was highly effective in a xenograft mouse model of the U-87 human glioblastoma cell line in vivo. We suggest that DR5-B-iRGD may become a promising candidate for targeted therapy for glioblastoma.

## 1. Introduction

The construction of bifunctional proteins is a powerful tool to enhance the efficiency of therapeutics by activating several signaling pathways with a single molecule. Bifunctional proteins, which simultaneously target alternative molecular pathways responsible for tumor development, are promising for cancer treatment. For example, bintrafusp alfa, the extracellular domain of the TGF-βRII receptor fused to a human IgG1 anti-PD-L1monoclonal antibody, is a first-in-class bifunctional fusion protein for dual targeting of TGF-β and PD-L1 pathways with multiple phase 2 clinical trials in solid tumors [1].

TRAIL (TNF-related apoptosis-inducing ligand) is a potential antitumor cytokine selectively targeting tumor cells without affecting normal cells. Despite the soluble domain of the wild type, TRAIL showed low antitumor efficiency in the first set of clinical trials [2]; its outstanding tumor specificity and safety encouraged the development of numerous strategies exploiting the TRAIL signaling pathway in tumor treatment. Along this path, numerous TRAIL-based bifunctional proteins were generated that specifically recognize various molecular targets for the destruction of tumor cells. For example, TRAIL fusions with single chain fragment variable (scFv) antibodies were developed, which target various tumor-specific antigens, which are often overexpressed in cancer, such as EGFR, ErbB2, Ep-CAM, or immune cell antigens, such as CD40, PD-L1, or IL2 receptors, to hasten and reinforce immune responses [3]. Moreover, TRAIL bispecific constructs selective for the annexin V, albumin binding domain (ABD), a small cationic linear α-helical peptide selective for tumor cells (CM4), Fn14, tumor molecular targeted peptide 1 (TMTP1), vasostatin have been described, all of which specifically bind to tumor cells and display increased TRAIL-mediated apoptosis [3].

Peptide iRGD (CRGDKGPDC) enhances tissue penetration by different molecules upon binding to integrins αVβ3 and αVβ5 followed by proteolytical cleavage and obtain specificity to the NRP-1 receptor [4]. This unique ability has been widely exploited to enhance the penetration of various compounds, from small molecules [5] to nanoparticles [6,7,8], liposomes [9], and exosomes [10,11]. The construction of chimeric proteins with the iRGD peptide was used to enhance the tumor permeability of the variable domain from the heavy chain of the anti-epidermal growth factor receptor antibody (anti-EGFR VHH) [12], recombinant analogue of lactaptin (RL2) [13], proapoptotic peptide KLA [14], interleukin-24 (IL-24) [15], and DNA fragmenting factor (DFF40) [16]. Moreover, fusion with iRGD increased binding to endothelial cells and improved the antiangiogenic property of endostatin [17], reduced the density in tumor vessels and accumulation in tumors of peptide hormone thymosin α1 (Tα1) [18], and improved the tumor tropism and parenchymal penetration of the anti-Tenascin-C antibody [19].

Moreover, the iRGD peptide was previously exploited to enhance TRAIL penetration into tumor tissue both in combination [20] and as a fusion molecule [21]. However, the aforementioned works used the wild type TRAIL, capable to bind both death receptors DR4 and DR5, as well as decoy receptors DcR1, DcR2, and OPG. This may cause a possible reduction in the antitumor effect by titration of the ligand by decoy receptors and activation of the anti-apoptotic signaling. 

Previously, we obtained receptor-selective TRAIL variant DR5-B specifically binding to the death receptor 5 (DR5) [22]. TRAIL variant DR5-B does not bind to decoy receptors, thus overcoming the potential receptor-dependent resistance of tumor cells to TRAIL [23]. However, this can be insufficient for the treatment of highly aggressive hard-to-treat types of solid tumors, such as glioblastoma. 

In the current work, we attempted to enhance the DR5-B tumor targeting and cytotoxicity by fusing it with the tumor-penetrating iRGD peptide. The antitumor activity of the obtained DR5-B-iRGD fusion protein has been investigated in human glioblastoma cell lines, primary patient-derived glioblastoma neurospheres, and the xenograft mouse model of the U-87 human glioblastoma cell line in vivo. In all cases, DR5-B-iRGD showed increased antitumor activity compared to DR5-B. Therefore, DR5-B-iRGD may become a promising candidate for the treatment of glioblastoma. 

## 2. Results

### 2.1. Expression and Purification of DR5-B-iRGD Fusion Protein

To obtain the genetic construct encoding fusion protein DR5-B-iRGD, the DNA sequence encoding the GGGGSGGGGSGG linker and the iRGD peptide (CRGDKGPDC) was inserted into the previously developed plasmid vector pET32a/*dr5-b* for the expression of the DR5-selective TRAIL (114–281) variant DR5-B protein [24] downstream the sequence encoding DR5-B. The schematic illustration of the fusion protein construct is shown in Figure 1A. The DR5-B-iRGD fusion protein was expressed by IPTG induction in *E. coli* strain SHuffle B and purified as described previously [24] (Figure 1B). A two-stage purification by affinity and ion-exchange chromatography allowed for obtaining a highly purified DR5-B-iRGD preparation (Figure 1B, lane 10). The yield of purified DR5-B-iRGD protein reached 48 ± 5 mg per 200 mL of the cell culture, which was twice as high as the yield of DR5-B (22 ± 4 mg) purified under similar conditions (Table 1) [25]. This indicates that fusion with the iRGD peptide may affect the DR5-B protein yield apparently by improving its solubility.

### 2.2. DR5-B-iRGD Showed High Affinity for DR5 and Integrin αvβ3

The affinity of the bispecific fusion protein DR5-B-iRGD to its target receptors DR5 and integrin αvβ3 was analyzed by an enzyme-linked immunosorbent assay (ELISA). The affinity of DR5-B-iRGD to the DR5 receptor (*K_D_* = 1.658 ± 0.080 nM) was similar to that of DR5-B (*K_D_* = 1.460 ± 0.066 nM). Therefore, the addition of an extra peptide iRGD to the C-end of the DR5-B protein did not affect its binding capacity to DR5 (Figure 2A,B). At the same time, DR5-B-iRGD bounded to the integrin αvβ3 in the nanomolar concentration range (*K_D_* = 1.886 ± 0.110 nM) (Figure 2C), which is an order of magnitude lower than the *K_D_* value reported for the free iRGD peptide (17.8 ± 8.6 nM) [4]. Expectedly, DR5-B alone did not show any apparent affinity for the integrin αvβ3 receptor (Figure 2C). In competitive ELISA, the free iRGD peptide displaced the DR5-B-iRGD upon binding to the integrin αvβ3 in a dose-dependent manner (Figure 2D), thereby demonstrating that the DR5-B-iRGD protein specifically binds to the same target site on the integrin αvβ3 as the original iRGD peptide. The affinity of DR5-B-iRGD to integrin αvβ5 was significantly lower (*K_D_* = 96,490 ± 11.341 nM); therefore, the integrin αvβ5 was not taken into account in the further experiments.

### 2.3. DR5-B–iRGD Was More Cytotoxic Than DR5-B in Human Glioblastoma Cell Lines T98G and U-87

The cell-death-inducing capacity of the DR5-B-iRGD protein was at first estimated in human glioblastoma cell lines T98G and U-87. Both cell lines expressed DR5 and integrin αvβ3 receptors on the cell surface, as it was shown by flow cytometry (Figure 3A). In T98G cells, the DR5-B-iRGD showed similar cytotoxic activity as DR5-B after a 24 h incubation, with the IC50 value of DR5-B-iRGD (0.436 ± 0.101 nM) slightly lower than that of DR5-B (0.530 ± 0.137 nM). During the further incubation for 48 h, there was no significant decrease in IC50 for none of the tested proteins (0.382 ± 0.081 nM and 0.445 ± 0.075 nM for DR5-B-iRGD and DR5-B, respectively) (Figure 3B,C). In U-87 cells, DR5-B-iRGD was twice as effective (IC50 = 1.112 ± 0.232 nM) as compared to DR5-B (IC50 = 2.206 ± 0.365 nM) at a short exposure period (24 h). With a longer exposure of cells to ligands (48 h), this difference increased to four-fold (IC50 = 0.253 ± 0.073 nM and 1.117 ± 0.261 nM for DR5-B-iRGD and DR5-B, respectively) (Figure 3B,C). Blocking the caspase-mediated apoptosis by pan-caspase inhibitor Z-VAD-FMK demonstrated that both DR5-B and DR5-B-iRGD induce cell death by a caspase-dependent mechanism (Figure 3D).

### 2.4. DR5-B-iRGD Demonstrated Enhanced Antitumor Effect in Primary Glioblastoma Patient-Derived Neurospheres

The main purpose of fusing the iRGD peptide to DR5-B protein was to enhance the drug penetration into tumor tissues. Therefore, the three-dimensional (3D) multicellular tumor structures can serve a better model for estimating the DR5-B-iRGD efficiency. In this regard, we tested the antitumor activity of DR5-B-iRGD in the previously obtained primary glioblastoma patient-derived cells (lines 11 and 267), which spontaneously aggregate into multicellular neurospheres [26]. Line 11 assigns to the proneural molecular subtype, while line 267 is of the mesenchymal subtype. Both of them expressed high amounts of DR5 and integrin αvβ3 receptors on the cell surface, as determined by flow cytometry (Figure 4A). DR5-B and DR5-B-iRGD demonstrated sufficient cytotoxicity in these cells, which was observable even by the morphology of the neurospheres (Figure 4D). Importantly, DR5-B-iRGD was significantly more cytotoxic than DR5-B in both 11 and 267 lines of the primary neurospheres (Figure 4B). The 267 line was particularly more sensitive to DR5-B-iRGD with IC50 = 0.006 ± 0.002 nM, compared with IC50 = 0.102 ± 0.028 nM for the DR5-B (Figure 4C). 

### 2.5. DR5-B-iRGD Penetrates into U-87 Spheroids Faster Than DR5-B

The ability of DR5-B-iRGD to penetrate into tumor structures was investigated on multicellular spheroids of the U-87 human glioblastoma cell line. For this, first the cysteine residue was introduced to the N-ends of DR5-B and DR5-B-iRGD protein sequences by site-directed mutagenesis, followed by labeling of the obtained proteins with fluorescent dye sulfo-Cyanine 3 maleimide by maleimide chemistry coupling. This ensured an equal ratio of the fluorescent dye molecules per DR5-B and DR5-B-iRGD ligands, and at the same time helped to avoid the possible hindering by the fluorescent dye molecules of specific binding of ligands to the DR5 and integrin αvβ3 receptors. Further, the U-87 multicellular spheroids were incubated with 5 nM of fluorescently labeled DR5-B or DR5-B-iRGD for the appropriate time periods, and the Z-stacks of the spheroids were visualized by laser scanning confocal microscopy. Expectedly, DR5-B-iRGD penetrated into U-87 spheroids faster compared with DR5-B: the fluorescence of DR5-B-iRGD was detected already after 15 min of incubation with the spheroids. On the contrary, the fluorescence of the DR5-B protein was slightly detected only after 30 min of incubation. The fluorescence intensities of DR5-B and DR5-B-iRGD became approximately equal only after 1 h of incubation (Figure 5). Together with the cytotoxicity studies, this indicates that DR5-B-iRGD obtains better tumor-penetrating and cell death-inducing capabilities in glioblastoma cells compared with DR5-B.

### 2.6. Comparative Analysis of DR5-B and DR5-B-iRGD Antitumor Activity in U-87 Human Glioblastoma Xenografts

The antitumor activities of DR5-B-iRGD and DR5-B were compared in vivo in subcutaneous U-87 human glioblastoma xenografts in BALB/c nu/nu mice. Mice (*n* = 5 per group) were inoculated with 1 × 10^7^ cells, and once the average size of the tumors reached 0.163 ± 0.032 cm^3^, vehicle, DR5-B or DR5-B-iRGD were injected intravenously at a dose of 10 mg/kg/d seven times every two days. The animals were monitored twice a week for 32 days. Both DR5-B and DR5-B-iRGD effectively reduced tumor volume (*p* < 0.0001) (Figure 6B). The tumor growth inhibition (TGI) by DR5-B-iRGD reached up to 82%, in contrast to up to 55% TGI by DR5-B (Figure 6C). The average TGI by DR5-B-iRGD was 1.6 times higher compared with TGI by DR5-B. Importantly, DR5-B-iRGD not only effectively reduced the tumor volume, but in two mice the complete resorption of the tumor nodes was observed (Figure 6A).

## 3. Discussion

Glioblastoma multiforme is a highly malignant primary brain tumor, which is considered resistant to wild type TRAIL [27]. However, recent encouraging studies with different compounds, which regulate the components of the TRAIL-dependent apoptotic signaling mechanism, show that targeting the TRAIL pathway is a promising therapeutic strategy for glioblastoma treatment [28]. Despite that several dozens of TRAIL-based fusion proteins were developed in the past few years, few of them are developed for brain tumor targeting, with the blood-brain barrier (BBB) being one of the important obstacles to the implementation of TRAIL-based preparations for the treatment of glioblastoma. To overcome this, an alternative virus-based therapeutic strategy was developed: the adenoviral vector expressing GFP-TRAIL fusion protein showed significant antitumor effects against intracranial xenografts of high-grade malignant meningioma and glioma [29]. However, the recently developed TRAIL-ANG2 fusion protein, which combines hexavalent TRAIL and angiopep-2 (ANG2), advanced the research by promoting receptor-mediated transcytosis across the BBB [30].

Previously, the researchers attempted to enhance TRAIL antitumor activity by functionalizing it with tumor-homing peptides bearing integrin recognition motif RGD (arginine–glycine–aspartic acid), particularly binding to integrins αvβ3/5, which are vastly expressed in most tumor cells and the tumor vasculature. For example, TRAIL fusion with the ACDCRGDCFC peptide (RGD-L-TRAIL) specifically bounded to microvascular endothelial cells and showed enhanced apoptosis-inducing activity in αvβ3 and αvβ5 integrin-positive cancer cells [31]. The RGD-TRAIL protein, in which TRAIL was fused with the RGD motif-containing cell adhesive sequence GRGDNP, possessed more potent antitumor effects than wild type TRAIL [32]. A tri-specific fusion protein RGD-TRAIL-NGR (ACDCRGDCFC-G4S-TRAIL-G4S-CNGRCVSGCAGRC) with motifs, specifically recognizing integrin ανβ3 and CD13, induced apoptosis in tumor cells and inhibited metastasis [33]. Most recently, a tumor-homing TRAIL fusion with the RGR (CRGRRST) peptide (RGR-TRAIL) showed enhanced cell binding and cytotoxicity in colorectal cancer cells and exerted enhanced tumor uptake and growth suppression of tumor xenografts compared with TRAIL [34].

In contrast to other tumor-targeting peptides with the integrin recognition motif RGD, the iRGD peptide (CRGDKGPDC) obtains the important property of additionally enhancing the drug penetration into tumor tissues. The mechanism includes the specific binding of iRGD to integrins αVβ3 or αVβ5, followed by a proteolytical cleavage and obtains specificity to the NRP-1 receptor, resulting in an NRP-1-dependent transcytosis [4]. Therefore, it is widely used as both a tumor-homing and a tumor-penetrating peptide for low molecules and macromolecular compounds [35,36]. Fusion with iRGD is of particular interest for targeting glioblastoma, since its unique properties allow for improving the penetration of various drugs through the BBB, which is one of the main obstacles to successful glioblastoma treatment. In this regard, iRGD was suggested as a promising candidate to overcome the poor permeability of therapeutics in the brain. Many studies have demonstrated that targeting with the iRGD peptide enhances the therapeutic outcomes in brain cancer treatment due to increased cell internalization and BBB penetration [37]. For example, the incorporation of the iRGD peptide resulted in an efficient drug delivery by the synthetic protein nanoparticles due to penetrating the BBB and distributing throughout the tumor volume [38]. Moreover, iRGD provided the high capacity of crossing the BBB, penetrating the tumor tissue, and accumulating in glioma for a theranostic nanodrug iRPPA@TMZ/MnO (an incorporated oleic acid-modified manganese oxide and temozolomide into polyethylene glycol-poly(2-(diisopropylamino)ethyl methacrylate-based polymeric micelles) [39].

Due to its dual tumor-homing and tumor-penetrating properties, genetic fusion with iRGD is used to enhance the cytotoxic activity of various antitumor proteins, including TRAIL. For example, the fusion protein TRAIL-iRGD demonstrated the enhanced antitumor effect against human gastric carcinoma xenografts [21]. In the current work, we generated the bispecific fusion protein DR5-B-iRGD based on the DR5-selective TRAIL variant DR5-B [22], which was aimed specifically to target two distinct receptors, DR5 and integrin αvβ3, which are both associated with tumor development. DR5 is traditionally considered as a more potent and crucial receptor for triggering TRAIL-dependent apoptosis [40]. Integrin αvβ3 is highly involved in tumor pathogenesis, being associated with the tumor growth and metastasis [41]. Despite that it was previously reported that the N-terminal rather than C-terminal chimeric protein fusions confer TRAIL-enhanced antitumor activity [42], while constructing the DR5-B-iRGD fusion protein, we placed the iRGD at the C-terminus of the DR5-B protein sequence to comply with the C-end rule, without which the iRGD peptide did not work [4]. Unexpectedly, the fusion of DR5-B with the iRGD peptide enhanced the target protein yield, presumably due to enhanced solubility. This correlates with the report that C-terminal fusion of an elastin-like polypeptide (ELP) to the RGD-TRAIL construct increased the solubility and yield of the target RGD-TRAIL-ELP protein [43]. Additionally, DR5-B-iRGD bounded to the integrin αvβ3 with affinity, which is an order of magnitude better than that reported for the free iRGD peptide [4]. 

The DR5-B-iRGD cytotoxicity was investigated in vitro not only in the established cell lines of human glioblastoma, but also in the primary patient-derived glioblastoma cells, which spontaneously aggregate into 3D multicellular neurospheres. The in vitro model of primary glioblastoma neurospheres better mimics the disease condition due to several advantages. Firstly, primary cells have not yet accumulated changes, which cells usually obtain during the linearization process, moving them away from their original disease state. Secondly, the three-dimensional structure allows better modeling of the tumor microenvironment, as well as the tumor-penetrating ability of the DR5-B-iRGD protein. We chose the primary cell line 11 of the proneural (PN) and line 267 of the mesenchymal (MES) subtype. Usually, tumors of the mesenchymal subtype are the most aggressive and resistant to therapy, while tumors of the proneuronal subtype are characterized by the best prognosis for patients [44]. However, given the established glioblastoma cell lines T98G and U-87, both lines of primary neurospheres were sensitive to the DR5-B and DR5-B-iRGD treatment. Importantly, the enhancement of the cytotoxic effect of DR5-B-iRGD compared with DR5-B was better pronounced in multicellular neurospheres than in the monolayer glioblastoma cell lines T98G and U-87. Presumably, this can be explained by the accelerated iRGD-induced penetration into their three-dimensional structures, which we further demonstrated on the U-87 spheroids. The increased antitumor effect of DR5-B-iRGD over DR5-B was also confirmed in vivo in the subcutaneous xenograft model of human glioblastoma cell line U-87 in nude mice. 

The research needs to be further advanced on an orthotopic intracranial human glioblastoma xenograft model to investigate the ability of the DR5-B-iRGD protein to penetrate through the BBB, accumulate in tumor tissue, and induce the reduction of tumor nodes. If it succeeds, DR5-B-iRGD may join the ranks of promising therapeutic agents against glioblastoma.

## 4. Materials and Methods

### 4.1. Reagents and Cell Lines

The ampicillin, IPTG (isopropyl-β-d-1-thiogalactopyranoside), WST-1 reagent, and CC/Mount tissue-mounting medium were obtained from Sigma-Aldrich (St. Louis, MO, USA); the Alamar Blue reagent was from Thermo Fisher Scientific (Waltham, MD, USA); the iRGD peptide was from InvivoChem (Libertyville, IL, USA); sulfo-Cyanine 3 maleimide was from Lumiprobe (Moscow, Russia); pan-caspase inhibitor Z-VAD-FMK was from Santa Cruz Biotechnology (Dallas, TX, United States). All other chemicals were obtained from Applichem (Darmstadt, Germany) unless otherwise specified. All solvents and components of buffer solutions were of analytical grade. Monoclonal antibodies to TRAIL (MAB375) were from R&D systems (Minneapolis, MN, USA); monoclonal antibodies to DR5 (DR5-01-1) and integrin αVβ3 (23C6), secondary antibodies Dylight 488 and mouse IgG1 (15H6) were from GeneTex (Irvine, CA, USA). *E. coli* SHuffle B T7 cells were from New England Biolabs (Ipswich, MA, USA. Bacterial cells were cultivated using Gibco Bacto yeast extract and Gibco Bacto tryptone (Thermo Fisher Scientific, Waltham, MA, USA). Human glioblastoma U87 and T98G cells were from ATCC (Washington, DC, USA). The cell culture media DMEM, 0.25% Trypsin-Versene solution, and phosphate-buffered saline tablets were from PanEco (Moscow, Russia). Thefetal bovine serum was from HyClone (Cramlington, UK).

### 4.2. Construction of Plasmid Vector for the Expression of DR5-B-iRGD Fusion Protein

The plasmid vector for the expression of the DR5-B-iRGD fusion protein was obtained by inserting the oligonucleotide sequence ggtggaggtggctcaggaggtggtgggagtggcggttgtcgtggtgacaaaggtcctgattgctaa, encoding the GGGGSGGGGSGG linker peptide and the CRGDKGPDC peptide, into the previously obtained plasmid vector pET32a/*dr5-b* [24] at the 3′ end of the DNA sequence encoding the DR5-B protein. The pET32a/*dr5-b-irgd* plasmid vector for the expression of the DR5-B-iRGD fusion protein was ordered at Evrogen (Moscow, Russia).

### 4.3. Expression of Recombinant DR5-B-iRGD Protein in E. coli 

For the expression of DR5-B and DR5-B-iRGD proteins, the competent cells of *E. coli* strain SHuffle B T7 were transformed with the plasmid vectors pET32a/*dr5-b* or pET32a/*dr5-b-irgd*. The colonies were inoculated into the LB (Lysogeny Broth, also called Luria–Bertani) medium in the presence of ampicillin (100 μg/mL), and cell cultures were grown at 37 °C with shaking at 250 rpm for 5 h. The cell culture was diluted (1:100) in TB (Terrific Broth) medium with ampicillin (100 μg/mL), grown at 37 °C to an optical density of OD600= 0.6, followed by adding 0.05 mM IPTG (isopropyl-β-D-1-thiogalactopyranoside). Cells were sedimented by centrifugation at 5000× *g* (Beckman Coulter, Indianapolis, IN, USA) at 4 °C for 10 min, then washed with a buffer containing 300 mM NaCl and 50 mM Na_2_HPO_4_ (pH 8.0), and the cell pellets were stored at −80 °C for further protein purification. The expression level of the target proteins was analyzed in 12% SDS-PAGE (sodium dodecyl sulfate polyacrylamide gel electrophoresis) with further staining by Coomassie Blue dye.

### 4.4. Purification of Recombinant Proteins DR5-B and DR5-B-iRGD 

The recombinant proteins DR5-B and DR5-B-iRGD were purified as described previously [24]. Briefly, cells were disrupted by a French press (Spectronic Instruments Inc., Irvine, CA, USA) under a pressure of 2000 psi, and the proteins were purified from the soluble fraction of the cytoplasmic proteins by immobilized metal-affinity chromatography on Ni-NTA agarose (Qiagen, Germantown, MD, USA), followed by ion exchange chromatography on SP Sepharose (GE Healthcare, Danderyd, Sweden). Purified protein preparations were dialyzed against 5 mM Na_2_HPO_4_ (pH 7.0) and 150 mM NaCl for 24 h at 4 °C, sterilized by filtration, lyophilized, and stored at −80 °C.

### 4.5. ELISA

The DR5 (100 ng/well) or integrin αvβ3 (200 ng/well) (R&D Systems, Minneapolis, MN, USA) receptors were immobilized on the Nunc MaxiSorp ELISA plates (Nunc, Roskilde, Denmark) overnight at 4 °C in 0.1 M carbonate–bicarbonate buffer (pH 9.4). The wells were washed three times with PBST (phosphate-buffered saline + 0.05% Tween), followed by blocking with 2% BSA in PBST for 1 h at 37 °C. Next, dilutions of DR5-B or DR5-B-iRGD (in 3 replicates) at concentrations from 0.032 to 50 or 500 nM in a dilution buffer (20 mM Tris-HCl, 150 mM NaCl, pH 7.3, 0.1% BSA, 1 mM CaCl_2_, 0.5 mM MgCl_2_) were added. For competitive ELISA, 100 μL of iRGD-peptide in dilution buffer were added with 2 nM of DR5-B-iRGD and incubated for 1 h at 37 °C. The captured ligands were detected by subsequent incubation with monoclonal antibodies to TRAIL (MAB375, R&D Systems, Minneapolis, MN, USA) and anti-mouse polyclonal goat horseradish peroxidase (HRP)-conjugated IgG (HAF007, R&D Systems). After washing 3 times with a PBST buffer, the color was developed by an OPD (o-phenylenediamine dihydrochloride) colorimetric substrate for 15 min at 37 °C. The reaction was stopped by a 1 N H_2_SO_4_ solution, and the optical density was determined at 450 nm by the iMark reader (Bio-Rad, Hercules, CA, USA). Dissociation constant (*K_D_*) values were calculated by GraphPad Prism 8 software (GraphPad Software Inc., San Diego, CA, USA) by the option of nonlinear regression in the XY analysis section.

### 4.6. Flow Cytometry

To determine the cell surface expression of DR5 and integrin αvβ3 receptors, the cells were seeded in the 6-well plates (2 × 10^5^ cells per well in 2 mL of culture media) and incubated at 37 °C and 5% CO_2_ for 24 h. After detachment from the culture flasks with Versene solution, the cells were washed with ice-cold PBS and resuspended in FACS buffer (PBS with 1% BSA), followed by incubation with 5 μg/mL of monoclonal antibodies to DR5 or integrin αvβ3 for 1 h at 4 °C. Further, the cells were washed twice and incubated with 20 μg/mL of the secondary antibodies Dylight 488 for 1 h at 4 °C, followed by washing twice and suspending in FACS buffer with propidium iodide. Mouse IgG1 was used as an isotype control. The expression of DR5 and integrin αvβ3 on the cell surface was measured on a CytoFlex flow cytometer (Beckman Coulter, Brea, IN, USA).

### 4.7. Cell Culture and Cell Viability Evaluation in Glioblastoma Cell Lines

U-87 and T98G human glioblastoma cell lines were cultured in DMEM supplemented with 10% FBS at 37 °C and 5% CO_2_. The cells were detached by a trypsin–EDTA solution (0.25% *v*/*v*), and the culture medium was replaced every 3–4 days. For cytotoxicity evaluation, the cells were seeded in 96-well plates (1 × 10^4^ cells per well) and incubated for 24 or 48 h with the dilutions of purified proteins DR5-B or DR5-B-iRGD. The colorimetric WST-1 assay was used for quantification of cell viability. The WST-1 solution was added into each well, and after 2 h incubation at 37 °C, the optical density was measured at a wavelength of 450 nm subtracting the background at 655 nm using an iMark Microplate Absorbance Reader (Bio-Rad, Hercules, CA, USA). The cell viability was determined by the percentage compared to the control according to the equation: (OD sample − OD background)/(OD control − OD background) × 100%. The half-maximal inhibitory concentration (IC50) was determined to be a drug concentration resulting in the 50% inhibition of cell growth by nonlinear regression in GraphPad Prism 8, (GraphPad Software Inc., San Diego, CA, USA) according to the built-in dose–response inhibition formula.

### 4.8. Cell Culture and Cell Viability Evaluation in Primary Glioblastoma Patient-Derived Neurospheres

Glioma tissue samples were obtained from the N.N. Burdenko National Medical Research Center of Neurosurgery (Moscow, Russia) and processed to the research laboratory after de-identification. Diagnosis was confirmed by morphological studies. The primary culture of glioblastoma cells as neurospheres was established as described previously [26]. Briefly, freshly resected GBM samples were dissociated, and neurosphere cultures were established and cultivated in DMEM/F12 medium containing 2% B27 supplement, 1% Penicillin-Streptomycin solution, 2.5 μg/mL heparin, 20 ng/mL basic fibroblast growth factor (bFGF), and 20 ng/mL epidermal growth factor (EGF). bFGF and EGF were added twice a week, and the cultural medium was changed every 7 days. The data with neurospheres were obtained with cells cultivated for no longer than 40 passages. The cell lines were tested negative for mycoplasma contamination. For cell viability evaluation, the cells were seeded in 96-well plates (6 × 10^3^ cells per well) and incubated for 72 h with the dilutions of purified proteins DR5-B or DR5-B-iRGD. Viability was evaluated by adding Alamar Blue reagent into each well, and, 6 h later, fluorescence was measured (Excitation 515–565 nm, Emission 570–610 nm) using a Synergy HTX multi-mode reader (BioTek, Winusky, VT, USA).

### 4.9. Fluorescent Labeling of DR5-B and DR5-B-iRGD Proteins

To obtain fluorescently labeled protein DR5-B-iRGD, first the DR5-B-iRGD amino acid sequence was genetically modified at the N-end by replacing the amino acid residue valine in the 114 position to cysteine by site-directed mutagenesis. The cysteine-modified proteins DR5-B and DR5-B-iRGD were obtained by the method earlier reported for DR5-B [45]. Further, the cysteine-modified proteins DR5-B and DR5-B-iRGD were labeled by maleimide chemistry coupling with the fluorescent dye sulfo-Cyanine 3 maleimide (Lumiprobe, Moscow, Russia) according to the manufacturer’s protocol.

### 4.10. Multicellular Tumor Spheroids Formation and Confocal Microscopy

Multicellular tumor spheroids were produced by the cell self-assembly method, as described earlier [46]. Briefly, U87 human glioblastoma cells were seeded in a 96-well plate (7500 cells/well) and incubated at 37 °C and 5% CO_2_ for 24 h until the cells attached to the plate bottom. Then, the medium was replaced in each well with 100 µL of complete DMEM (10% FBS) containing cyclo-RGDfK(TPP) peptide (40 µM) and incubated at 37 °C and 5% CO_2_. Spheroid formation with an average size of 100 ± 20 µm was observed in 72 h. The obtained spheroids were incubated with Hoechst 33258 (10 µg/mL in PBS) for 20 min, and further washed three times with PBS. Next, 50 nM of DR5-B or DR5-B-iRGD proteins were added and incubated for appropriate time periods followed by washing three times with PBS. The spheroids were transferred to the coverslips and fixed with the CC/Mount mounting medium. The samples were visualized by the inverted microscope Nikon TE-2000 configured with the C1 confocal system (Nikon, Tokyo, Japan). 

### 4.11. Xenograft Studies

Specific pathogen-free (SPF) BALB/c nu/nu, female, 8–12-week-old mice were obtained from the Russian National Center for Genetic Resources of Laboratory Animals at the Institute of Cytology and Genetics, Siberian Branch, Russian Academy of Sciences. All animals were admitted with a veterinary passport and certificate of quality. All procedures for routine animal care were performed in accordance with standard operational procedures and the sanitary rules for the design, equipment, and maintenance of experimental biological clinics and the Laboratory Animals manual. The antitumor effect of DR5-B and DR5-B-iRGD in vivo was assessed on a human glioblastoma xenograft model in BALB/c nu/nu nude mice. U-87 cells (1 × 10^7^ per mouse) in the BD Matrigel Basement Membrane Matrix (BD Biosciences, Franklin Lakes, NJ, USA) were inoculated subcutaneously in the right dorsal flank of 8–12-week-old mice. Tumors were developed in the absence of treatment until they were well established. The DR5-B, DR5-B-iRGD, and vehicle were administered intravenously in the tail vein. Once tumors reached a volume of 0.163 ± 0.032 cm^3^, mice were randomly divided into groups. The DR5-B or DR5-B-iRGD were administered to U-87 xenografts in a single dose of 10 mg/kg/d seven times every two days. Control animals were injected with an isotonic (0.9%) solution of sodium chloride (Solopharm, St. Petersburg, Russia) in a volume equivalent to the volume of the administered preparation. During the observation time, the general condition and animal weight were monitored. Tumor volumes were measured twice a week by percutaneous measurement of tumor formations using an electronic digital caliper STORMTM 3C301 «Central» in three mutually perpendicular projections and calculated by the formula V = π/6 × (width)^2^ × (length). Values of tumor growth inhibition were calculated by the formula [(Vc − Vex)/Vc] × 100%, where Vex and Vc are the tumor volumes in experimental and control groups. In animals with tumors, a decision concerning euthanasia was based on the assessment of the general condition, measuring the tumor size, critical loss in body weight, and the severity of tumor necrosis.

### 4.12. Statistical Analysis

ELISA and cell culture experiments were repeated at least three times independently. GraphPad Prism 8.0 (GraphPad Software Inc., San Diego, CA, USA) for Windows was used to generate graphical representations and for statistical analysis. The data were normally distributed and displayed as the mean ± SD from at least three replicates. In cell viability experiments, statistical analysis was performed using Student’s t-test or one-way ANOVA followed by either Dunnett’s or Šidák’s post hoc tests. In xenografts studies, the normal distribution of tumor volume data was checked by the Shapiro–Wilk test using GraphPad Prism 8; *p* < 0.0001 indicates a significant difference between data groups according to the two-way ANOVA test. The statistical analysis of tumor growth inhibition was performed by Student’s t-test. 

## 5. Conclusions

We generated a novel TRAIL-based fusion protein DR5-B-iRGD for tumor targeting and enhanced penetration into three-dimensional multicellular tumor structures. The obtained bispecific protein exhibited high affinity to DR5 and integrin αvβ3 receptors. DR5-B-iRGD demonstrated an enhanced antitumor effect in vitro in the established cell lines of human glioblastoma T98G and U-87 and in primary glioblastoma patient-derived neurospheres, and in vivo in a xenograft mouse model of human glioblastoma cells U-87. After further evaluation of the ability of DR5-B-iRGD to penetrate through the BBB and antitumor activity on orthotopic intracranial human glioblastoma xenografts, DR5-B-iRGD can become a potential candidate for glioblastoma treatment. 

## Figures and Tables

**Figure 1 ijms-23-12687-f001:**
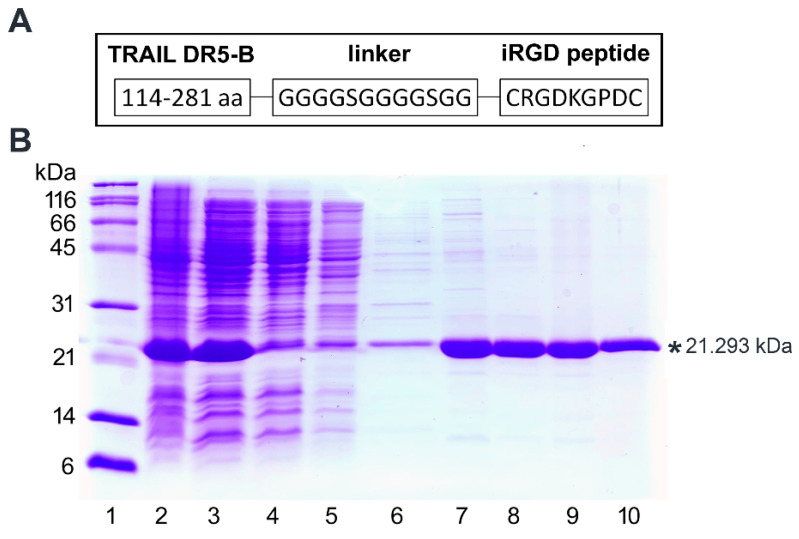
Expression and purification of DR5-B–iRGD fusion protein. (**A**) Scheme of the DR5-B–iRGD fusion construct. (**B**) Expression and purification of DR5-B–iRGD protein from the cytoplasmic cell fraction. Samples (10–15 µg protein per lane) were analyzed on SDS–PAGE. Lane 1—molecular weight markers; Lane 2—expression of DR5-B-iRGD in *E. coli* SHuffle B strain; Lane 3— soluble cell fraction isolated from bacterial lysate; Lanes 4–7—purification by metal affinity chromatography on Ni–NTA agarose: Lane 4—fraction proteins not associated with Ni–NTA sorbent; Lanes 5, 6—fraction of proteins washed by buffer with 10 mM imidazole; Lane 7—protein fraction eluted by a buffer with 250 mM imidazole; Lane 8—protein sample after overnight dialysis; Lane 9—purification by ion exchange chromatography on the SP Sepharose; Lane 10—purified protein after overnight dialysis. *—calculated mass of the monomeric DR5-B-iRGD protein 21.293 kDa.

**Figure 2 ijms-23-12687-f002:**
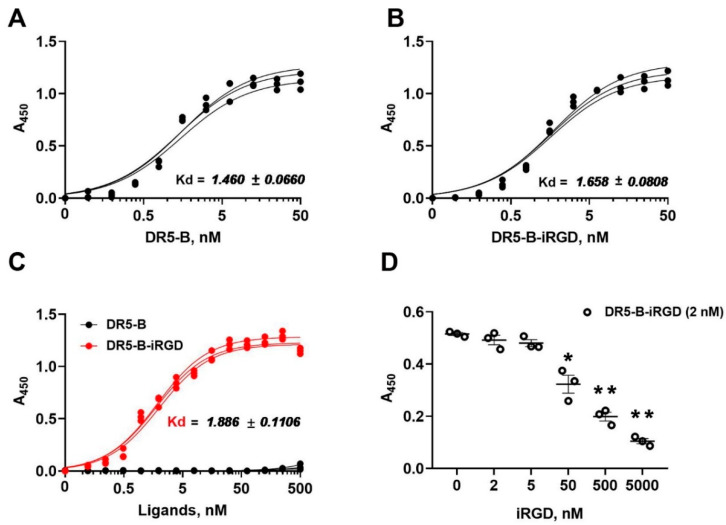
The binding affinity of DR5-B-iRGD protein for receptors DR5 and integrin αvβ3 analyzed by enzyme-linked immunosorbent assay (ELISA). (**A**) Affinity of DR5-B for the DR5 receptor. (**B**) Affinity of DR5-B-iRGD for the DR5 receptor. Significant difference between affinities of DR5-B and DR5-B-iRGD to DR5 receptor (*p* = 0.0303) was observed according to Student’s t-test. (**C**) Comparison of the affinities of DR5-B and DR5-B-iRGD for the integrin αvβ3. (**D**) Competitive binding of DR5-B-iRGD protein (2 nM) and free iRGD peptide to integrin αvβ3. * *p* < 0.005 and ** *p* < 0.001 indicate significant difference from the control according to one-way ANOVA followed by Dunnett’s post hoc test. *K_D_* values were calculated from three independent experiments by the non-linear regression option in GraphPad Prism 8.0.

**Figure 3 ijms-23-12687-f003:**
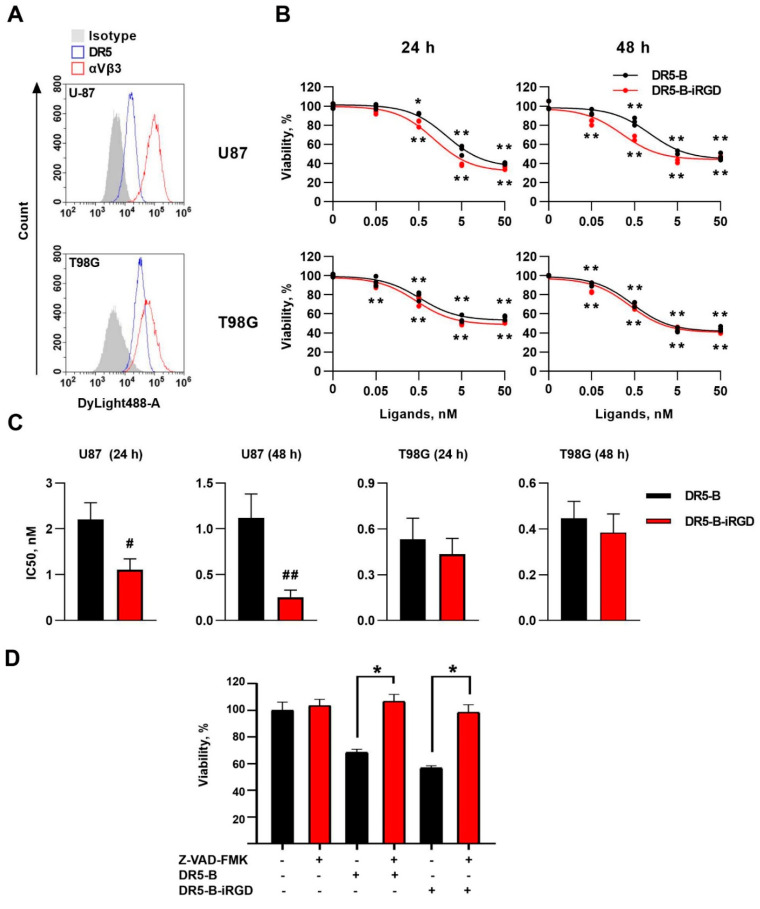
Cytotoxicity of DR5-B and DR5-B-iRGD in human glioblastoma cell lines U-87 and T98G. (**A**) Surface expression of the DR5 and integrin αvβ3 receptors determined by flow cytometry. (**B**) T98G and U-87 cells were incubated with DR5-B or DR5-B–iRGD for 24 or 48 h, and viability was analyzed by the WST-1 assay. The data were displayed as mean ± SD from at least three replicates. * *p* < 0.05 and ** *p* < 0.005 indicate significant difference from the control according to one-way ANOVA followed by Dunnett’s post hoc test. (**C**) IC50 values of DR5-B and DR5-B-iRGD were determined as the drug concentrations resulting in the 50% inhibition of cell growth by nonlinear regression in GraphPad Prism 8 software. ^#^ *p* < 0.05 and ^##^ *p* < 0.005 indicate significant difference from the control according to Student’s t-test. (**D**) Cells were incubated with 20 µM Z-VAD-FMK for 1 h followed by treatment with 5 nM of the ligands for 24 h.* *p* < 0.0001 indicates significant difference from the cells not treated with Z-VAD-FMK according to one-way ANOVA followed by Šidák’s post hoc test.

**Figure 4 ijms-23-12687-f004:**
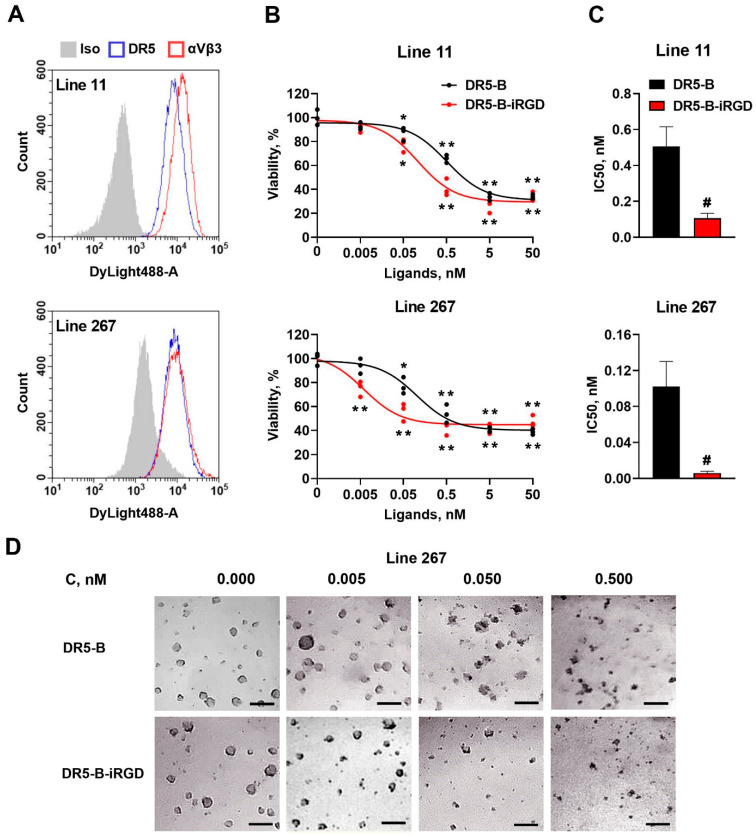
Cytotoxicity of DR5-B and DR5-B–iRGD in primary human glioblastoma patient-derived neurospheres. (**A**) Surface expression of the DR5 and integrin αvβ3 receptors determined by flow cytometry. (**B**) Primary human glioblastoma cell lines 11 and 267 were incubated with DR5-B or DR5-B–iRGD for 72 h, and viability was analyzed by the Alamar Blue assay. The data were displayed as mean ± SD from at least three replicates. * *p* < 0.005 and ** *p* < 0.0005 indicate significant difference from the control according to one-way ANOVA followed by Dunnett’s post hoc test. (**C**) IC50 values of DR5-B and DR5-B–iRGD were determined as the drug concentrations resulting in the 50% inhibition of cell growth by nonlinear regression in GraphPad Prism 8 software. # *p* < 0.005 indicates significant difference from the control according to Student’s *t*-test. (**D**) Morphology of primary patient-derived neurospheres (Line 267) treated with DR5-B or DR5-B-iRGD. Scale bar is 200 µm.

**Figure 5 ijms-23-12687-f005:**
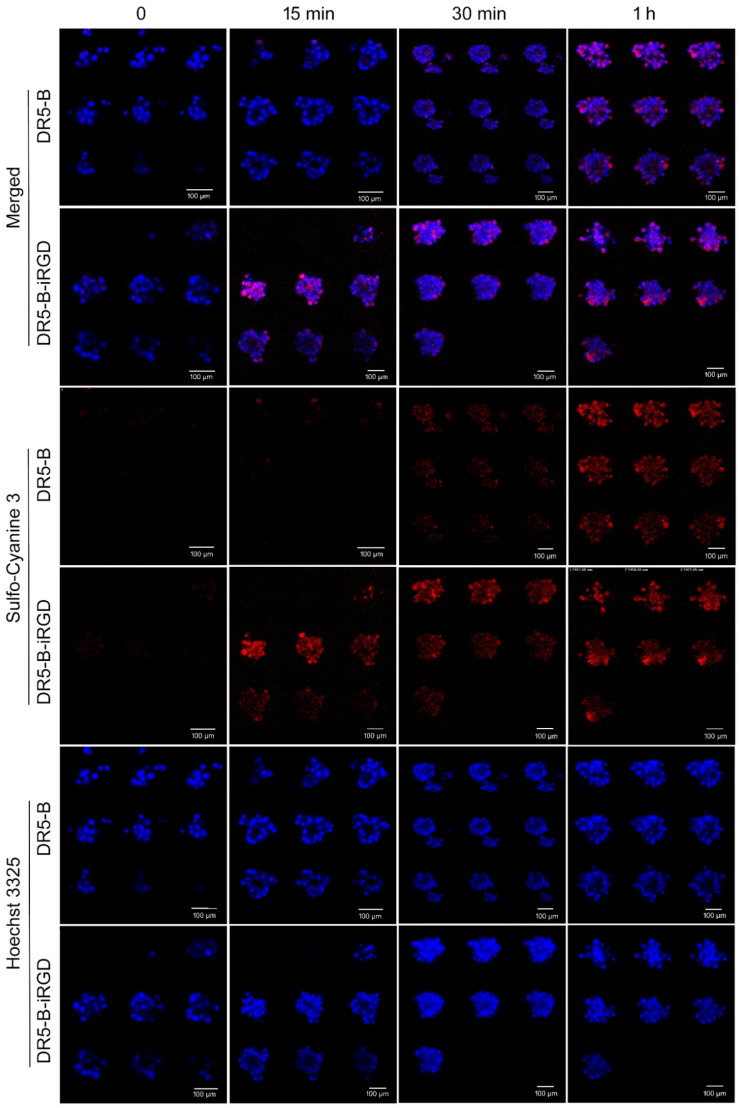
Penetration of the DR5-B and DR5-B-iRGD into multicellular tumor spheroids of U-87 human glioblastoma cell line. Confocal laser scanning microscopy Z-stack images of U-87 human glioblastoma multicellular spheroids after 15 min, 30 min, and 1 h incubation with 5 μM of DR5-B or DR5-B-iRGD. DR5-B and DR5-B-iRGD are in red (sulfo-Cy3); cell nuclei are in blue (Hoechst 33258). Scale bar is 100 μm; a step for Z-stack images is 5–15 μm.

**Figure 6 ijms-23-12687-f006:**
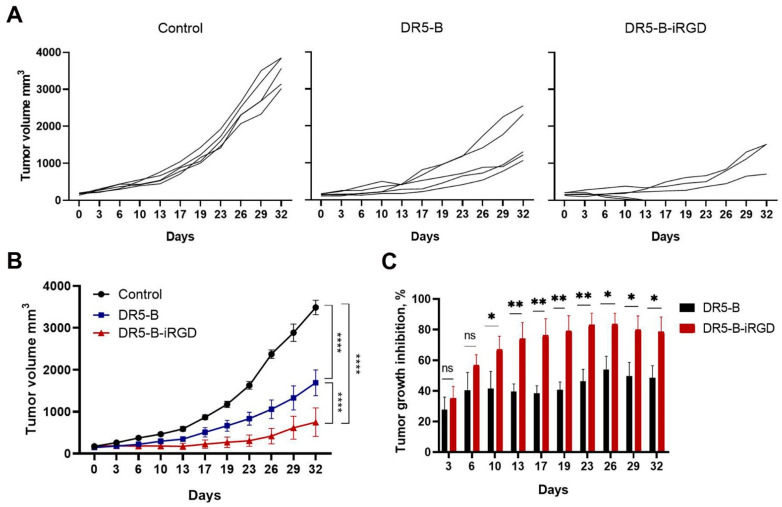
Comparison of DR5-B and DR5-B-iRGD antitumor activity in vivo. U-87 human glioblastoma xenograft tumors were established in Balb/c nu/nu mice by subcutaneous injection of 1×10^7^ cells. When tumor volume reached 0.163 ± 0.032 cm^3^, mice were treated by intravenous injection of 10 mg/kg/d of preparations (seven times every two days). Arrows show the scheme of drug administration. (**A**) Tumor growth curves for individual mice in each treatment group (*n* = 5). (**B**) Tumor volumes were determined every 3–4 days by percutaneous measurement with caliper. Tumor volumes presented as mm^3^ ± SEM (*n* = 5). **** (*p* < 0.0001) indicates significant difference between data groups according to two-way ANOVA test. (**C**) Values of tumor growth inhibition were calculated by the formula [(Vc − Vex)/Vc] × 100%, where Vex and Vc are the tumor volumes in experimental and control groups (*n* = 5). * (*p* < 0.05) and ** (*p* < 0.005) indicate significant difference according to Student’s t-test.

**Table 1 ijms-23-12687-t001:** Yields of DR5-B and DR5-B-iRGD proteins from 200 mL liter of cell culture during purification.

Purification Step	DR5-B, mg	DR5-B-iRGD, mg
Soluble fraction of cytoplasmic proteins *	350 ± 18	480 ± 23
Purification on Ni-NTA agarose *	98 ± 7	134 ± 11
Purification on SP Sepharose **	22 ± 4	48 ± 5

* Amount of protein determined by the Bradford method. ** Amount of protein determined by absorption at 280 nm regarding the calculated extinction coefficient.

## Data Availability

All data generated or analyzed during this study are included in this article.

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
