# Peer review of "DR5-Selective TRAIL Variant DR5-B Functionalized with Tumor-Penetrating iRGD Peptide for Enhanced Antitumor Activity against Glioblastoma"

_ijms, 2022, doi:10.3390/ijms232012687_

Round 1

Reviewer 1 Report

The present manuscript established the potential of a newly-developed bifunctional protein based on 1) a TRAIL sequence specific to DR5 and 2) an iRGD peptide allowing better penetration into tumor tissues, for the treatment of glioblastoma.

The hypothesis developed in this study is presented in a consistent way, results are displayed very clearly. The conclusions are not pretentious but straightforward, practical and useful for the researchers in the field of TRAIL-based anticancer therapies.

A few suggestions for further improvement are listed below :

- Is the cell morphology/adherence modified upon treatment with DR5B-iRGD ? And as for the neurospheres, do they form as efficiently as in basal conditions ? Representative images would support the data.

- Signaling experiments could be helpful to show the extent of TRAIL-induced signaling downstream DR5, upon DR5B or DR5B-iRGD binding.

- The U87 model is well-accepted as not ideal, which is a major weakness. Could the authors take advantage of the patient-derived GSCs for the study of DR5B-iRGD penetration in vitro ? And for in vivo experiments ?

- Could the authors verify and compare the penetration of both DR5B and DR5B-iRGD in the subcutaneous tumor tissue (if conserved after the experiment) ?

Author Response

We appreciate the positive evaluation of our work. We have made the following adjustments in accordance with the comments.

- Is the cell morphology/adherence modified upon treatment with DR5B-iRGD? And as for the neurospheres, do they form as efficiently as in basal conditions? Representative images would support the data.

Response: Upon DR5-B-iRGD treatment, change in morphology of monolayer cells U-87 and T98G was similar to that upon DR5-B treatment. The primary neurospheres have been pre-formed in basal conditions before treatment, and subsequently underwent changes characteristic of cell death upon further incubation with ligands. Thank you for your suggestion, we have included the additional figure 4D to support this data.

- Signaling experiments could be helpful to show the extent of TRAIL-induced signaling downstream DR5, upon DR5B or DR5B-iRGD binding.

Response: Thank you for your remark. We have conducted additional experiment with pan-caspase inhibitor Z-VAD-FMK, which blocks the features of apoptosis, to demonstrate that both DR5-B and DR5-B-iRGD induce cell death by caspase-dependent mechanism. The corresponding figure 3D is added to the manuscript.

- The U87 model is well-accepted as not ideal, which is a major weakness. Could the authors take advantage of the patient-derived GSCs for the study of DR5B-iRGD penetration in vitro ? And for in vivo experiments?

Response: We understand the limitations of the applied U87 model. However, the aim of the current pilot study was to obtain the bispecific fusion protein DR5-B-iRGD and to preliminary confirm its functional activity. Undoubtedly, now when we have evaluated its potential as an antitumor protein with improved characteristics, we will advance the research by applying more relevant in vitro and in vivo models, including the orthotopical xenografts of patient-derived GSCs. 

- Could the authors verify and compare the penetration of both DR5B and DR5B-iRGD in the subcutaneous tumor tissue (if conserved after the experiment)?

Response: Unfortunately, the subcutaneous tumor tissues have not been conserved after the experiment. In further work, we are going to compare the penetration of fluorescently-labelled DR5-B and DR5-B-iRGD proteins through the blood-brain barrier and into the orthothopically established intracranial glioblastoma xenografts in vivo.  

Reviewer 2 Report

This manuscript by Isakova et al aims to improve the antitumor activity of DR5-B by fusion with a tumor-homing iRGD peptide. Purified DR5-B-iRGD fusion protein showed high affinity for DR5 and Integrin αvβ3 and was found to be more cytotoxic than DR5-B in human glioblastoma cell Lines T98G and U-87. Moreover, DR5-B-iRGD was found to penetrate into U-87 Spheroids faster than DR5-B and demonstrated enhanced antitumor effect in primary glioblastoma patient-derived neurospheres. Finally, DR5-B-iRGD effectively reduced tumor volume in vivo.

Overall, this is a well-executed study which will be helpful to develop novel strategies for targeting glioblastoma. I accept the manuscript in the present form for publication.

Author Response

Thank you, we appreciate the positive evaluation of our work.

Author Response

Major points

-There is representation of the statistical analysis for figure 2, figure 3B and 3C, figure 4B and 4C and 6C. Then, authors cannot assume that DR5b-iRGD fusion protein has a higher affinity to the receptors as well as a higher cytotoxic profile and antitumor activity. The statistical analysis performed has to be described in the text of each figure, as well as level of statistical significance (p).

Response: Thank you for your comment. We have introduced the description of the statistical analysis and the level of statistical significance to the captions of the aforementioned figures. As for figure 6C, it does not include statistical analysis; the values of tumor growth inhibition (TGI) were calculated from the averaged tumor volumes for experimental and control group (n=5). We have mentioned this in the figure caption.

-The number of replicates is not indicated for each figure.

Response: We have indicated the number of replicates in the captions of figures, where needed.

-Authors should include a western-blot image for demonstrating DR5 and alfa-V-beta3 expression in T98G and U87 cell lines, as well as in primary glioblastoma patient-derived cells (lines 11 and 267).

Response: As far as DR5-B-iRGD is a bifunctional ligand, which activates the downstream signaling upon binding to its target receptors on the cell surface, the surface expression level of these receptors is much more important than the total expression. Thus, we considered the flow cytometry data more informative than that obtained from western blot. 

-A complementary assay should be included to demonstrate cytotoxicity of DR5b-iRGD as the measure of lactate dehydrogenase release (LDH assay).

Response: Thank you for your suggestion. Actually, in the current study we have applied the most relevant assays for monitoring the cytotoxicity, which are highly informative and reliable. The WST-1 assay is optimal due to its water solubility and superior sensitivity; it lacks toxicity and allows omit additional washing and harvesting steps. The same applies even more to the Alamar Blue assay, a highly sensitive a direct indicator of cell health, which detects the level of oxidation during respiration in real-time. Importantly, LDH assay was shown to be not optimal for detection of low cytotoxic effects as well as effects at low compound concentrations due to low signal-to-noise ratio and the high inter- and intra-assay variabilities (https://doi.org/10.1186/1471-2210-8-8). Therefore, we consider that additional measurements of the number of dead cells by quantifying LDH levels will be redundant.

-Authors should include more that 2 primary glioblastoma patient-derived cells lines to demonstrate cytotoxicity of DR5b-iRGD fusion protein.

Response: We understand that more cell lines are needed to evaluate reliably the antitumor potential of DR5-B-iRGD protein. However, in the current pilot study we intended to report the production, stability and functional activity of the newly developed bispecific protein DR5-B-iRGD. The studies are going to be continued on the broad spectrum of cell lines of various origin as well as extended panel of primary glioblastoma patient-derived cells.

-It will be very representative to include a picture of tumour volume in xenografts after different treatments.

Response: Unfortunately, this item was omitted in our study. We appreciate your remark and we will definitely take it into account in future work.

Minor points

-Figure 2A and 2B could be mixed in only one graph similar to figure 2C

Response: Originally we were going to plot the data exactly as you describe. However, as a result, the curves and individual points overlapped each other and the graph became less informative, so we decided to split the data into two graphs.

-Include between lines 134 and 142 that “DR5-B-iRGD bound to the integrin .v.3 receptor with high affinity that DR5-B alone”

Response: Thank you for your suggestion. Actually, DR5-B alone expectedly did not show any apparent affinity for integrin αvβ3 receptor. We have placed this statement at the specified location in the text.

-line 100. from the cytoplasmic cell could be modified by “bacteria”

Response: The indicated text has been modified.

-indicate that figure 1 has been obtained after Coomassie staining in “mat and methods section”

Response: The “Materials and methods” section has been supplemented with the information about gel staining.

Round 2

Reviewer 1 Report

The authors addressed most of the suggestions and comments, and improved the quality of the work. I therefore recommend the Editors to accept the present manuscript for publication in IJMS.

Author Response

We are grateful for the positive evaluation of our work.

Reviewer 3 Report

-Why statistical analysis is not represented in figure 6C? 

-line 380. "Cumassie" is wrong should be corrected by "Coomassie"

-line 100 has not included "bacteria" instead of cytoplasmic cell.

Author Response

-Why statistical analysis is not represented in figure 6C? 

Response: Originally, the values of tumor growth inhibition (TGI) were calculated from the averaged tumor volumes for experimental and control groups. For statistical analysis to be applicable, we recalculated the TGI values for individual animals in each group and performed the statistical analysis by Student’s t-test using GraphPad Prism 8 software.

-line 380. "Cumassie" is wrong should be corrected by "Coomassie"

Response: The misprint has been corrected.

-line 100 has not included "bacteria" instead of cytoplasmic cell.

Response: Actually, Lane 3 shows a sample of the soluble cell fraction isolated from bacterial lysate by centrifugation. We have modified the caption to Figure 1, Lane 3, to “soluble cell fraction isolated from bacterial lysate”.
